# Data Center Traffic Prediction Algorithms and Resource Scheduling

**DOI:** 10.3390/s22207893

**Published:** 2022-10-17

**Authors:** Min Tan, Ruixuan Ba, Guohui Li

**Affiliations:** 1School of Computer Science and Technology, Huazhong University of Science and Technology, Wuhan 430074, China; 2China Academy of Information and Communications Technology, Beijing 100191, China; 3School of Computer Science and Technology, Wuhan University, Wuhan 430072, China

**Keywords:** tree–seed optimizer algorithm, LSTM model, data center, load balancing scheduling

## Abstract

This paper uses intelligent methods such as a time recurrent neural network to predict network traffic, mainly to solve the problems of resource imbalance and demand differentiation under the current 5G cloud-network collaborative architecture. An improved tree species optimization algorithm is proposed to optimize the initial network data, and the LSTM model is used to predict the data center traffic to obtain better network traffic prediction accuracy, take corresponding measures, and finally build a scheduling algorithm that integrates business cooperative caching and load balancing based on traffic prediction to reduce the peak pressure of the 5G data center network.

## 1. Introduction

With the advent of the 5G era, the global digital industry is accelerating. As the core infrastructure for digital transformation and upgrade, 5G data centers play an underlying role in supporting the digital economy. 5G applications are gradually being released and exploded. In the first half of 2021, the operator’s network disconnection accident rate increased for the first time in nearly five years, and network overload operation has become the norm [1]. At present, there are widespread pain points of unbalanced resource allocation in 5G networks, low fault location efficiency, and high energy consumption, which have brought great challenges to the operation and maintenance of 5G data centers [2].

The “post-epidemic era” has accelerated the development of digital technology, and the application and integration of industrial digitalization has continued to deepen. According to the authoritative forecast of the China Academy of Information and Communications Technology, China’s digital economy will exceed 60 trillion by 2025, and the digital industrialization proportion will reach more than 10%. By 2025, my country, China, will build a full-coverage 5G network, and the scale of direct investment will exceed 2 trillion yuan, which will support the scale of industrial internet and intelligence manufacturing by more than 600 billion yuan. In 2020, my country proposed to “accelerate the progress of new infrastructure construction of the 5G network and data center network”. The commercial use of the 5G network is in full swing, and the multi-polar evolution of network services “cloud, management, edge and end” will be the mainstream trend in the future. At this stage, 5G data centers and network equipment generally suffer from unbalanced resource utilization, low operation and maintenance efficiency, and high energy consumption [3].

The 5G data center is the core infrastructure for the development of “new infrastructure” and the digital base for the development of my country’s digital economy. It is estimated that the total investment in my country’s network operation and maintenance will exceed 100 billion in 2021, and the industry demand is strong. At present, the operation and maintenance of data centers mainly rely on manual inspection and other methods. However, with the gradual commercialization of 5G and the explosive growth of application scenarios, the pressure on network load cannot be resolved. At the same time, under the new development pattern of carbon peaking and carbon neutrality, the data center, as the space carrier of digital transformation, is also an inevitable trend of my country’s construction of industrial green and low-carbon industrial pattern transformation.

In this paper, we propose a scheduling algorithm that integrates business cooperative caching and load balancing based on traffic prediction. The rest of this article is organized as follows. In Section 2, related research trends on traffic forecasting are explained. Section 3 proposed Tree–Seed Optimizer Algorithm, Attention Mechanism and AM-based LSTM prediction model. Section 4 provides business scheduling algorithm. The Section 5 analyses the experimental results and proves the validity of the model. Finally, the Section 6 summarizes this paper.

## 2. Related Works

The current architecture of data centers is evolving toward centralization and marginalization. In comparison to large data centers, the energy-saving technologies and strategies of small and medium-sized data centers are far behind those of large data centers. The cost pressure of commercial and operating enterprises is huge [4,5]. At present, the industry mainly studies the dynamic environment of the data center, but the energy saving effect is still not significant. In the context of the dual carbon strategy, the problem of how to effectively reduce the energy consumption of the data center will become a challenge that must be overcome in digital transformation [6].

The primary task of traffic forecasting on communication network equipment is data collection and analysis. In the past, most of the time, professional analysts collected and organized data in the early stages and conducted detailed analysis of the data based on experience to give comprehensive judgments and propose network optimization. However, there is a shortage of resources for senior technical experts in my country, and it is not realistic to simply test human resources to organize and analyze complex network data. References [1,7] indicate that many scholars and experts in traffic forecasting technology have conducted a lot of scientific research on network traffic forecasting and have achieved many gratifying scientific research results and proposed many new basic models of network traffic forecasting [8,9,10]. At present, the methods of forecasting network traffic mainly include artificial intelligence forecasting methods and time series forecasting methods, both of which are essentially involved in the learning and forecasting of time series data. The artificial intelligence prediction method can process massive amounts of information, has the advantages of a high error tolerance rate for sample data, strong nonlinear mapping ability, an adaptive learning ability and high operating efficiency, which makes it possible to fit more complex nonlinear features.

Traffic forecasting refers to a process of making accurate judgments on a certain changing law. In recent years, with the rapid development of artificial intelligence technology, traffic forecasting has played an important role in all walks of life. At present, the three major operators are actively exploring research on traffic forecasting. Among them, China Mobile has carried out the most extensive research [11]. China Mobile took the lead in proposing “industry digitization and intelligent upgrade, becoming a pioneer and enabler of artificial intelligence applications”, and actively researching the “Jiutian” artificial intelligence platform has established its leading position among operators in artificial intelligence research [12]; China Telecom proposed the transformation goal of “network intelligence, business ecology, and operation intelligence”, and released the “Lighthouse” artificial intelligence platform [13,14,15]; China Unicom put forward the goal and vision of “becoming a promoter of smart application development, a builder of smart networks, and a leader in smart technology innovation” [16]. Operators have put forward long-term goals for intelligent network traffic forecasting, which will promote the rapid progress and development of this industry and will also become a new force in the upgrading and transformation of 5G networks.

At present, some work has applied deep learning technology to network traffic prediction, and to some extent, it has achieved little success, but there are still some limitations in some aspects. For example, the data collection and analysis, the determination of the prediction model structure and the determination of super parameters [17], etc., all of which greatly affect the accuracy of the final traffic prediction of the model. Therefore, in order to obtain more satisfactory prediction results, this project proposes the use of intelligent optimization algorithms to optimize the data processing methods and the super parameters in the deep learning model, and the use of better processed data combined with the optimized deep learning model to predict the traffic of different network devices in real life, so as to obtain better network traffic prediction accuracy. Prejudgment and corresponding measures can be made in advance before problems such as congestion occur.

## 3. Proposed System

Using intelligent methods such as a time recurrent neural network to predict network service traffic in small and medium-sized data centers, it aims to reduce communication costs and improve resource utilization through accurate traffic prediction and dynamic adjustment of data center content distribution and vertical applications. It also aims to solve outstanding problems such as resource imbalance and demand differentiation under the current 5G cloud-network collaborative architecture, build traffic forecasting, build a scheduling algorithm that integrates business collaborative caching and load balancing, and reduce peak pressure on data center networks through traffic scheduling.

### 3.1. Tree–Seed Optimizer Algorithm

In order to solve the continuous optimization problem, Kiran first proposed the tree-seed algorithm (TSA). Compared with some traditional intelligent optimization algorithms, the tree-seed algorithm has a simpler structure, more accurate search degree, and stronger robustness. In this paper, the tree species optimization algorithm is improved, and an improved tree species optimization strategy is proposed based on the flight strategy. Feature selection is performed on the network data set, and the features of irrelevant or redundant data sets are eliminated to improve the accuracy of the network data set [9].

The pseudo code of the algorithm is summarized in Algorithm 1.
**Algorithm 1** Lévy Flight Improved Tree–Seed Algorithm (LTSA)1. Initialization algorithmGenerate N random trees in the D-dimensional search space by using the formula; Calculate the position of the tree according to the problem objective function; Choose the optimal solution according to the formula.2. Search for torrentsDetermine the number of seeds this tree produces; Each dimension of the FOR seed; Update this dimension according to the formula; Choose the best seed and compare it with this lesson tree; If the seed is in a better position than the tree, replace the tree with the seed.3. Select the optimal solution in the population If the newly generated optimal solution is better than the previous optimal solution, the new optimal solution and its corresponding tree are generated.; Or replace the original optimal solution and the corresponding tree.4. Check whether the iteration termination condition is meet If not satisfied, go back to Step2.5. Output the optimal solution

### 3.2. Attention Mechanism

The birth of the Attention Mechanism (AM) comes from the human visual attention mechanism, which is essentially the response of the human brain to external signals. When humans observe things, they always allocate more attention to a specific area. They pay attention, and then obtain more feature information from this area, while observing other areas at a lower resolution, and the corresponding amount of information obtained is relatively small. Attention will have different perceptions of external things, and there will be probability distributions. At the same time, important parts will occupy more weight in the calculation. Through this feature, the important parts will be more prominent, and the model can also be better and have an excellent effect. The key to the attention mechanism is to configure weights for important information, making the model more sensitive to important information and improving the efficiency and accuracy of information processing. The steps for the attention mechanism to calculate the attention variable are shown in Figure 1.

The values copied Tx times using the RepeatVector node in the above figure are then used to calculate the sum connection using Concatenate, which is then calculated by the activation softmax function of the output layer, and finally used to calculate the output variable for each time step.

### 3.3. AM-Based LSTM Prediction Model

Based on the idea of Attention, applying it to the time series prediction model of LSTM can also be used to calculate the different influences of the prediction results on the input data. Compared with the traditional method, it can highlight the key points and has a stronger ability to mine the potential relationship of the sequence, so as to optimize the time series feature vector and improve the prediction performance, as follows:

(1) Using the Encoder–Decoder model, the main purpose is to input a serialized data, and after the encoding and decoding process, the output is also a serialized data. The encoding process is shown in the figure as the intermediate sequence obtained by the input data through the cyclic calculation of the LSTM unit, and the decoding process is shown as a similar sequence model, and the new output information is predicted through the historical output information.

(2) Attention mechanism based on weight distribution. Its impact factor is in the former LSTM layer (Encoder part) and the latter LSTM layer (Decoder part), the weight will be assigned according to the important key of the network data characteristics, and the network traffic time series will be predicted at the same time. Live capture. 

(3) Use the Attention mechanism to complete the multiple calculations of the LSTM unit until the output information of the short-term network traffic and the characteristic information of its influencing factors after the integration of the attention mechanism are obtained, which can satisfy the input information of the prediction layer.

The prediction framework based on the AM-LSTM model is shown in Figure 2.

Here are some key properties of the model:

(1) Definitions of the two LSTM models in Figure 2. The LSTM model before the attention mechanism is defined as pre-attention LSTM in this paper. The LSTM model behind the attention mechanism is defined in this paper as post-attention LSTM. Pre-attention LSTM is over Tx time steps, while post-attention LSTM is over Ty time steps.

(2) In this paper, the RNN neural network model will be applied in the post-activation sequence to obtain the output state s(t) of the RNN neural network model. Moreover, due to the use of LSTM results in this paper, the long short-term memory model LSTM has both the output activation state s(t) and the hidden unit state c(t). However, unlike the previous ones, in this model, post-activation LSTM will not generate a specific y(t-1) as input at time t; it only needs to generate s(t) and c(t) as input.

### 3.4. AM-LSTM Model Training Steps

The training steps of the AM-LSTM short-term network traffic prediction model proposed in this paper are as Algorithm 2.
**Algorithm 2** Long Short-Term Memory Model (AM-LSTM) Introducing Attention Mechanism**Input:** Network traffic and its impact factor training set and test set.**Output:** The network traffic value at a certain moment.Complete the forward propagation calculation of the prediction model at the AMLSTM network layer;1. Initialize the parameters to be learned in AMLSTM;2. Introduce the attention mechanism (AM) into the LSTM neural network layer. Multiple memory units use the attention mechanism to calculate the probability distribution of the output features multiple times, and reweigh the output features as the input of post-LSTM;3. Backpropagation is performed in the LSTM neural network layer. At this time, the error value of each neuron on the hidden layer of the LSTM network needs to be calculated, and the partial derivative is obtained, and the corresponding weight and bias are updated;4. Iterative training to the set number of iterations, ending the model parameter training;5. Test the model, and judge whether the model is good or bad through the evaluation indicators;6. Back-normalize the output data with the time series Y test data and compare it with the real value to evaluate the predictive ability of the model.

## 4. Business Scheduling Algorithm

### 4.1. Business Collaborative Cache and Load Balancing 

First, in order to overcome the challenges brought about by the coupling relationship between service caching and workload scheduling, the computation-transmission delay trade-off, and the heterogeneity of edge nodes, this scheme proposes service caching and workload scheduling strategies.
(1)$$C=\left(c_{ns}\in \left\{0,\;1\right\}:n\in N,\;s\in S\right)$$
(2)$$\Lambda =\left(\lambda_{ns}\in \left[0,\;1\right]:n\in N\cup \left\{o\right\},\;s\in S\right)$$

Among them, it indicates whether the business has been cached in the micro data center network and indicates the business workload ratio executed in the micro data center network. The service cached on the network of each micro data center cannot exceed its storage capacity, that is, the storage capacity required to represent the cached service. In addition, it needs to be satisfied.

Second, determine the average response time of the business, which is the weighted sum of the delays of the various parts of the system:(3)$$D_{s}=\sum_{n\in N}\left[\lambda_{ns}D_{ns}+\frac{\max\left\{\lambda_{ns}A_{s}-A_{ns},0\right\}}{A_{s}}\right]+\lambda_{os}d_{\mathrm{cloud}}$$

Among them, $\lambda_{ns}D_{ns}$ represents the computing delay of the micro data center network. $\frac{\max\left\{\lambda_{ns}A_{s}-A_{ns},\;0\right\}}{A_{s}}$ represents the transmission delay and $\lambda_{os}D_{\mathrm{cloud}}$ represents the transmission delay to the cloud data center network.

When selecting a micro data center network for collaborative computing, based on the above traffic prediction results, a micro data center network with a lower load is preferentially selected to reduce the peak pressure of a single micro data center network.

Finally, in order to minimize the service request response time and the traffic load of the cloud data center network, the above problem is transformed into a solution:(4) C, Λmin∑s∈SDs+ωsλosAs
(5)$$s.t.\;\underset{\;n\in N\cup \left\{o\right\}}{\sum }\lambda_{ns}=1,\;\;s\in S$$

Among them, $\omega_{s}$ is the weight constant, which is proportional to the network data flow. This paper adopts a two-layer iterative cache update algorithm, the outer layer iteratively updates the network data caching strategy based on Gibbs sampling (service caching sub-problem), and the inner layer optimizes the workload scheduling strategy (workload scheduling sub-problem).

In order to balance the calculation and transmission delay, this paper uses the queuing model to analyze the delay of each part of the system, so as to achieve a business balance between the calculation speed and the transmission delay.

### 4.2. Multi-Priority Queue Cache Management Method

Before giving the specific algorithm, the following definitions are given:

**Definition** **l.***Cache pre-allocation time interval*T=t2−tl. *Where*t1*is the pre-allocation time of the current cache, and*t2*is the pre-allocation time of the next cache*.

T reflects the frequency of cache pre-allocation, and the smaller T is, the more frequent the cache pre-allocation is.

**Definition** **2.***The average flow of data streams with priority*λ*in the future T time is*Lλ.

**Definition** **3.**Suppose there are n priority queues inside the node, then the cache pre-allocation coefficients of the n priority queues are δ1, δ2, ⋯, δn. Among them δ1: δ2:⋯ :δn=L1:L2:⋯: Ln.

The algorithm is divided into two parts: (1) cache allocation at the pre-allocation time; (2) cache adjustment and competition within time T. In the following description, the priority of digging a priority queue is sequentially decreased from n to 1, which is no longer specified.

The algorithm is described in detail as follows:

Initialization: the cache size of *n* priority queues S1, S2,⋯, Sn is cleared; let  j=1, k=1;

1. When the cache pre-allocation time is reached, the cache is allocated to each priority queue according to the pre-allocation coefficient:
(6)$$S_{k}=S*\delta_{i}/\left(\delta_{1}+\delta_{2}+\cdots +\delta_{n}\right),\left(i=1,2,\cdots ,n\right)$$

S is the sum of the internal cache of the node;

2. If a data packet arrives, go to (3), otherwise go to (1);

3. Read the priority i of the data packet, check whether the corresponding i priority queue has enough cache. If the cache is enough, insert the queue, and go to (1);

4. If j≤n go to (5), otherwise go to (6);

5. If the j-priority queue has an unused buffer, take out a fixed-size block of △ and assign it to the i-priority queue, while Sj=Sj−△,Si=Si+△, and insert the data packet into the i-priority queue, let j=1 turn (1); otherwise j=j+1, turn (4);

6. If k<i go to (7), otherwise go to (8);

7. If $Sk/S>\delta k/\left(\delta 1+\delta 2+\cdots +\delta n\right)$, take a fixed-size buffer from the k priority queue △ and assign it to the i priority queue, and at the same time Sk=Sk−δ, Si=Si+△, and insert the data packet into the i priority queue, let k=1 turn (1); otherwise k=k+1, turn (6);

8. Discard the packet and go to (1).

Through this algorithm, it can be seen that when a data packet (priority i) arrives, as long as there are idle cache resources inside the node, no matter which priority queue the cache is currently in, it can be used for the required queue (such as Algorithm (4), (5) two steps). When the cache resources inside the node are not in an idle state, search from the low priority queue in turn and compare with the pre-allocation ratio to identify whether the low priority queue is occupying more cache and make reasonable allocations. However, this operation is limited to low-priority queues, and has no effect on high-priority queues. If the high-priority queue occupies more cache resources (such as steps (6) and (7) of the algorithm), after reaching the pre-allocation time, the cache resources should be re-cached according to the average network traffic in the future T time. Therefore, it will not cause malicious preemption of cache resources required by low-priority queues, thereby ensuring the fairness of system resources.

It can be seen from the above description that the algorithm can not only ensure the priority of high-priority data streams, but also take into account the transmission of low-priority data streams and can fully utilize cache resources.

### 4.3. Load Balancing Scheduling Method

Load balancing is one of the important means to improve network performance. Realizing load balancing through some methods of machine learning can improve the performance of communication networks. Traffic transfer is one of the methods to achieve network load balancing in communication networks. It is the core mechanism of load balancing. Through traffic scheduling, services in heavy load areas are allocated to light load areas, so that traffic is distributed evenly throughout the communication network to the greatest extent. Among them, the load balancing model based on the traffic prediction algorithm can be used to improve the utilization of communication network resources.

This algorithm mainly involves Weighted Least-Connection Scheduling (WLCS for short), which is a dynamic load balancing algorithm. The basic idea of the algorithm is as follows: Suppose the set of network nodes is $S=\left\{S1,S2,S3,\cdots ,Sn\right\}\;(n>1)$ and the weight of each network node set in this set is $W\left(Si\right)\;(1<i\leq n)$, the corresponding number of connections established by each network node set is $L\left(Si\right)\;(1<i\leq n)$, then the connection of the total network node set is shown by the following formula:(7)$$M=L(S_{1})+L(S_{2})+\ldots +L(S_{n})$$

If the *i*-th server is represented as:(8)$$\frac{L(S_{i})}{W(S_{i})}=\min\left\{\frac{L(S_{j})}{C}/W(S_{j})\right\},(1\leq i\leq n,1\leq j\leq n)$$

The above formula indicates that the ratio of the number of connections of the nodes in the network node set to its weight is the smallest. If a new request is sent, the request will be sent to the network node with the smallest ratio. Where C represents a non-zero constant, the above formula can be simplified to:(9)$$\frac{L(S_{i})}{W(S_{i})}=\min\{L(S_{j})/W(S_{j})\},(1\leq i\leq n,1\leq j\leq n)$$

Since the weights of server nodes are all positive numbers, the above formula is simplified to:(10)$$\frac{L(S_{i})}{W(S_{i})}\leq \frac{L(S_{j})}{W(S_{j})}$$

Multiplication requires fewer CPU cycles than division, so division can be converted to multiplication. Convert the above formula to obtain the formula:(11)$$L(S_{i})*W(S_{j})\leq L(S_{j})*W(S_{i})$$

According to the idea of the weighted least connection scheduling algorithm, as long as the conditions of the above formula are satisfied, traffic resources can be preferentially allocated to network nodes.

## 5. Experiments and Results

The technologies and products developed in this project are mainly used in communication networks and have been tested in more than three local actual networks. The results meet the expectations, and it is planned to support operators to carry out network secondary expansion and optimization. The relevant data are the data packets collected in the actual transmission net for two months.

### 5.1. Preparation before Experiment

The flow forecasting system in this paper is divided into different subsystems as a whole. According to the logical association, it is divided into: the application front end, microservice resource layer, and business layer. The experiments settings and technical selection are shown in Table 1 and Table 2, respectively.

Business layers include:

1. System panel: This interface is used to set system parameters. Currently, the network name parameter can be set.

2. File archiving panel: This interface is used to set the monitoring folder for file archiving. After setting, the program will monitor the update of any file in the folder and archive the data in real time. Folders are relative locations based on the project root.

3. Data extraction panel: This interface is used to set the setting information required for data extraction processing. The data type drop-down list contains the data type of the performance indicator. The flow characteristic drop-down list and the data drop-down list are linked to display the flow characteristic indicators of the selected data type. They work together to locate a specific feature data of a specific data type.

4. Data cleaning panel: This interface is used to set the information required for data cleaning processing. The Delete All Empty Data switch is used to turn the deletion of all empty data in the data cleaning process on or off. The vacancy rate threshold input box is used to set the minimum data vacancy rate that meets the valid data.

5. Data enhancement panel: This interface is used to set the information required for data enhancement processing. The Simple Linear Interpolation switch is used to turn simple linear interpolation on or off. The Execute Lagrangian Interpolation switch is used to turn the text box of the number of points near the Lagrangian reference on or off and to turn Lagrangian interpolation on or off.

6. Cluster analysis panel: This interface is used to set the information required for cluster analysis processing. This function is set because the aggregation network element and the access king element are mixed together in the data, which will affect the corresponding elements’ prediction performance. The data cluster drop-down list will display the data volume of the two network elements obtained by the K-Means clustering analysis function. The K-Means cluster analysis execution button is used to perform cluster analysis on data.

The artificial intelligence module is developed in Python 3.8, based on the TensorFlow 2 framework, and the API uses Keras. Data processing uses NumPy and Pandas modules, and graph generation uses Matplotlib and Plotly modules.

The AI business process includes nine stages: data evaluation, data preprocessing, model building, parameter and hyperparameter optimization, training model, evaluating model, delivering model, forecasting, and generating charts.

The relevant parts are described as Table 3.

### 5.2. Traffic Prediction Experiment Results

This paper proposes the use of intelligent optimization algorithms to optimize data processing methods and hyperparameters in deep learning models, the use of better processed data, and combination of optimized deep learning models for data center traffic forecasting to obtain better network traffic forecasts accuracy. This experiment predicts the traffic of different time lengths such as 1 day, 4 days, and 7 days in the future, and the prediction accuracy rate is above 70%. The traffic forecast results are shown in Figure 3.

The prediction results panel of this experiment is used to display the prediction results. In this interface, firstly enter the node keyword, query the candidate list of a network element according to the keyword, fill in the candidate node drop-down list, and display the result by selecting the Wang Yuan node in the drop-down list. By clicking the PERV and NEXT buttons, it is used to display the data of the switching result between the NE of the previous serial number and the NE of the next serial number. The results are shown in Figure 4.

Lines showing two colors in the prediction result panel respectively represent: green as the real data; red as the predicted data. The experimental results of this paper have been used in multiple networks in Hubei for actual tests and have achieved good results in the trend prediction of network traffic. The analysis of the network traffic prediction algorithm model is helpful for operators in network operation and maintenance. Discover hidden dangers in business and dynamically adjust network resources in time, make predictions in advance and take corresponding measures before network failures, congestion and other problems occur, effectively improving network service quality.

### 5.3. Analysis of the Results of Resource Optimization Experiments

In this paper, intelligent methods such as a time recurrent neural network are used to predict network traffic, and an optimization model for business cooperative caching is constructed based on the traffic prediction results. The purpose is to reduce the communication cost and resource utilization of 5G network content distribution and vertical applications. The experimental results show that the resource balancing ability of the system has been significantly improved, and the resource utilization rate has dropped. The results are shown in Figure 5.

## 6. Conclusions

The main problems of data center network operation and maintenance are inefficient: the monitoring of core devices scattered in the network consumes too many communication resources, the abnormal detection cycle is too long, and the resource allocation strategy is inefficient, making it difficult to meet the application requirements of multiple types of services. In the future, the network application volume of 5G data centers will be large, which requires efficient systems, equipment operation and maintenance methods, and resource allocation strategies. To this end, this paper adopts a variety of intelligent methods such as machine learning and deep learning to improve the operation, maintenance efficiency and autonomy of 5G data centers.

With the rapid development of the economy and the ever-increasing updating to communication technology, the scale of my country’s communication network continues to expand. The adoption of a large number of advanced technologies has brought about problems such as an increasing number of network devices, complex and diverse types of communication services, diverse application scenarios, difficulty locating faults, and network traffic congestion. In the face of increasing traffic congestion, in order to improve the efficiency of network bandwidth allocation, provide an analysis basis for network traffic, routing selection, fault location, etc., reduce current operating costs, and increase subsequent operating income, it is necessary to strengthen the analysis of network equipment and routines. It is necessary to continuously adjust and optimize the layout and structure of network equipment in reality to meet the needs of users’ daily network traffic while reducing maintenance costs. However, in real life, the traditional operation and maintenance method based on post-event manual analysis and positioning has gradually been unable to meet the needs of daily network governance, and the advantages of pre-event intelligent prediction and analysis technology have become more obvious. With the integration of artificial intelligence, big data and other technologies, the intelligent analysis technology represented by deep learning has made great progress in network data processing, which has made new breakthroughs in network traffic prediction technology.

This paper uses historical network traffic data collected by 5G data center network equipment over a long time period to predict the value and trend of network traffic in a short period of time in the future and then deploys network topology change services in advance according to the prediction results, thereby reducing services due to equipment overload. loss. At the same time, it can provide data references and a basis for network topology optimization services and reduce network construction costs.

## Figures and Tables

**Figure 1 sensors-22-07893-f001:**
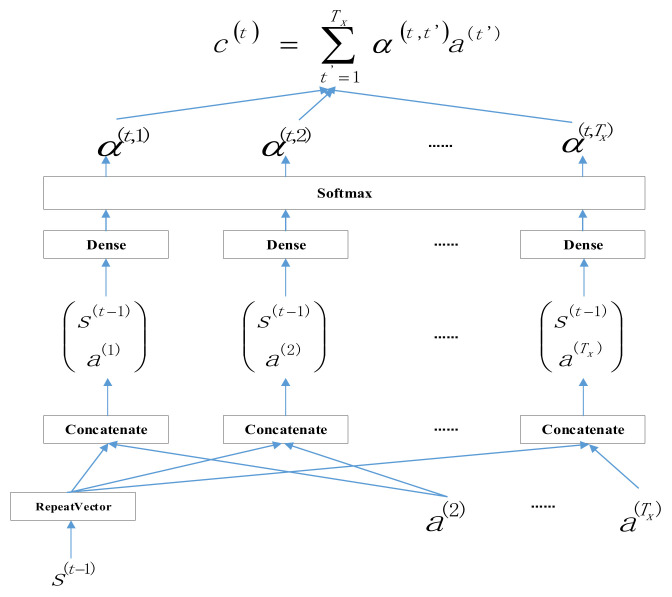
Attention mechanism.

**Figure 2 sensors-22-07893-f002:**
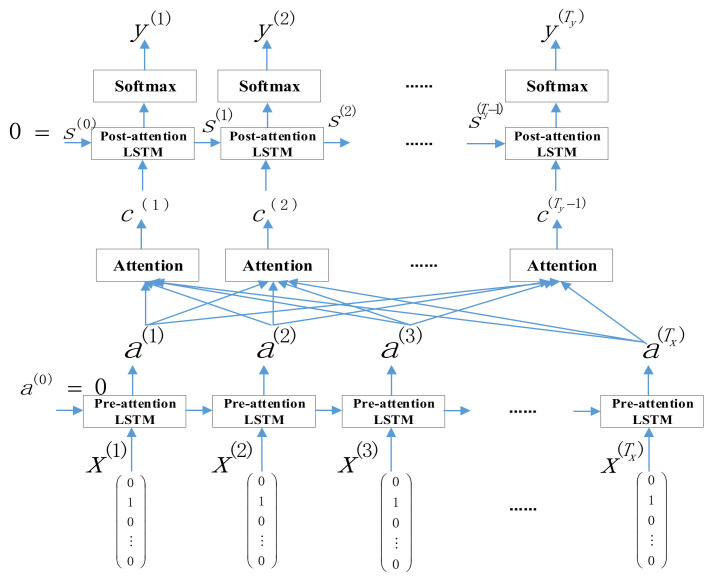
AM-LSTM model.

**Figure 3 sensors-22-07893-f003:**
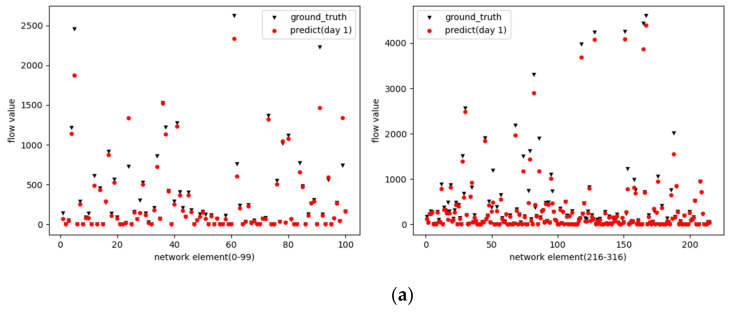
The traffic forecast results. (**a**) 1 day forecast results. (**b**) 4 days forecast results. (**c**) 7 days forecast results.

**Figure 4 sensors-22-07893-f004:**
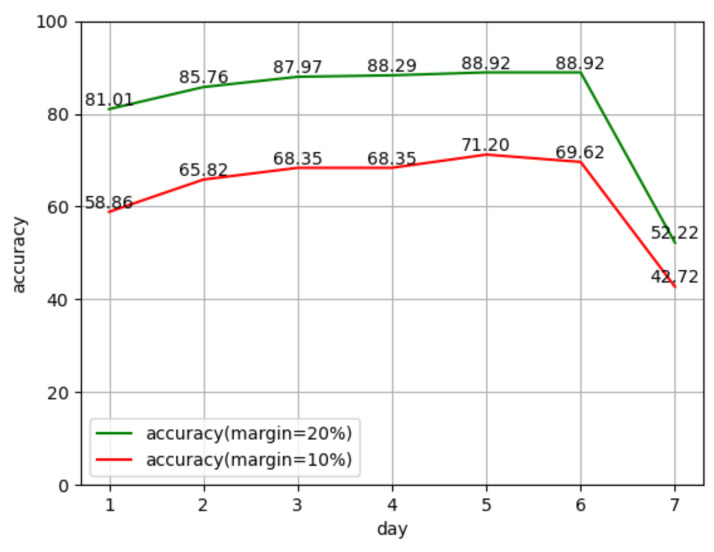
Prediction Results Panel Results.

**Figure 5 sensors-22-07893-f005:**
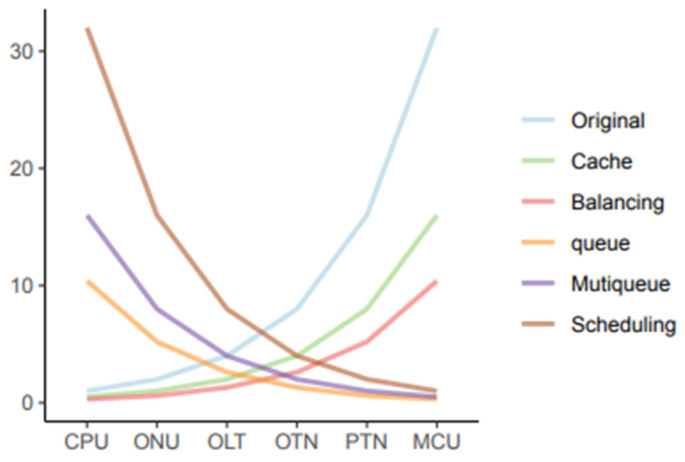
Resource optimization experiment results.

**Table 1 sensors-22-07893-t001:** The experiments settings.

Type	Settings
Processor	No less than 10 cores
Many-core general-purpose computing processor	Single-precision floating-point performance not less than 10 Tera FLOPS
Memory	No less than 64 GB
Client minimum	WebKit or Chromium-based browsers

**Table 2 sensors-22-07893-t002:** Technical selection table.

Type	Technical Selection
Operating system	Windows, Linux
Application server	Flask-Server
Database	MySQL 8
Library management	Anaconda
Integrated Development Environment	PyCharm
Data processing framework	NumPy, Pandas
AI development framework	TensorFlow 2
AI development interface	Keras
Persistence	openpyxl, SQLAlchemy

**Table 3 sensors-22-07893-t003:** Related Parts Instruction Sheet.

Part Name	Parts Description
Web front	Asynchronous update interface developed based on JQuery and Semantic UI
System Settings Module	Set system parameters
data archiving module	Set data archiving parameters, mainly monitor folders, and perform data archiving
data extraction module	Select data types and traffic characteristics, and perform data extraction
Data cleaning module	Set data cleaning parameters and perform data cleaning
Data Enhancement Module	Set data augmentation parameters and perform data augmentation
Cluster Analysis Module	Perform cluster analysis, giving category candidates
Model training module	Set model training hyperparameters and perform training
Forecast display module	Graphical display of evaluation value, actual value, and predicted value

## Data Availability

Not applicable.

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
