# Peer review of "Data Center Traffic Prediction Algorithms and Resource Scheduling"

_sensors, 2022, doi:10.3390/s22207893_

Round 1

Reviewer 1 Report

This paper discusses network traffic prediction problems for 5G data centres. This is a very important problem of huge relevance to society and industry. The authors could include a section before sec. 3 to mathematically formulate the scheduling problem that they are trying to solve without skipping any details of the elements involved in the real world operations of the data centres that they are trying to study. The algorithm is described well but the 3 layers of the forecasting system, viz. Application front end, micorservice resource, and business layer is not described with clarity. If the authors do not wish to describe fully because of space constraints they might at least cite se good resource where reader can pick up the missing details. The metrics used to study the prediction accuracy must be made more sophisticated by referring to existing machine learning literature.

Author Response

Thanks for your kind advice, please see the attachment.

Reviewer 2 Report

The paper in current state cannot be considered for publication, yet there are some changes could enhance the quality of the paper.

·         Introduction section without references, which is I recommend the authors to add some of them.

·         At the of introduction section, I recommend the authors to add paper outlines.

·         In related works section the authors should introduce some limits for the presented works.

·         In algorithm 1 and 2, the authors should present it as listing in table (please see MDPI template or previous published papers).

·         In sub-section 3.1 the authors referred in the paragraph “…, Kiran first proposed the tree- seed algorithm (TSA). I was expecting a reference at the end of the sentence.

·         Equation presented in subsection 4.1 are not well presented, the authors should unify the writing font style, they should also present the significance of used variables in each equation.

·         Some references are too old 2012, 2013 I recommend the authors to renew the references.

·         The authors should introduce a good presentation of their proposed methodology.

·         The authors should also give some details about the used data.

Author Response

Thanks for your valuable advice, please see the attachment.

Reviewer 3 Report

Overall, this paper is easy to follow, with a clear definition of the research problem and detailed explanations of the technological processes. The experiments were well designed, and the results are convincing. Thus, I would recommend accepting this paper.

Some suggestions for improving the paper:

1. It is better to use a more formal form to present the algorithms in P3 and P7.

2. Section 5.1 could be improved. A table is suggested to summarize the experimental settings.

3. The fonts used in figures can be enlarged, making the figures clearer.

Author Response

(The authors gave the same response as above.)

Round 2

Reviewer 2 Report

The required remarks has been carried out by the authors.